# Infrapatellar Fat Pad Modulates Osteoarthritis-Associated Cytokine and MMP Expression in Human Articular Chondrocytes

**DOI:** 10.3390/cells12242850

**Published:** 2023-12-15

**Authors:** Ewa Wisniewska, Dominik Laue, Jacob Spinnen, Leonard Kuhrt, Benjamin Kohl, Patricia Bußmann, Carola Meier, Gundula Schulze-Tanzil, Wolfgang Ertel, Michal Jagielski

**Affiliations:** 1Department of Traumatology and Reconstructive Surgery, Campus Benjamin Franklin, Charité—Universitätsmedizin Berlin, Corporate Member of Freie Universität Berlin, Humboldt-Universität zu Berlin and Berlin Institute of Health, Hindenburgdamm 30, 12203 Berlin, Germany; ewa.wisniewska@charite.de (E.W.); dominik.laue@charite.de (D.L.); jacob.spinnen@charite.de (J.S.); leonard.kuhrt@charite.de (L.K.); benjamin.kohl@charite.de (B.K.); patricia.bussmann@charite.de (P.B.); carola.meier@charite.de (C.M.); wolfgang.ertel@charite.de (W.E.); 2Institute of Anatomy and Cell Biology, Paracelsus Medical University (PMU), Prof.-Ernst Nathan Strasse 1, 90419 Nuremberg, Germany; gundula.schulze@pmu.ac.at

**Keywords:** IPFP, OA, osteoarthritis, cartilage, chondrocytes, MMPs

## Abstract

Osteoarthritis (OA) most frequently affects the knee joint and is associated with an elevated expression of cytokines and extracellular cartilage matrix (ECM), degrading enzymes such as matrix metalloproteinases (MMPs). Differences in gene expression of the intra-articularly located infrapatellar fat pad (IPFP) and other fatty tissue suggest its autonomous function, yet its role in OA pathogenesis remains unknown. Human IPFPs and articular cartilage were collected from OA patients undergoing total knee arthroplasty, and biopsies from the IPFP of healthy patients harvested during knee arthroscopy served as controls (CO). Isolated chondrocytes were co-cultured with either osteoarthritic (OA) or CO-IPFPs in a transwell system. Chondrocyte expression of *MMP1*, -*3*, -*13*, *type 1* and *2 collagens*, *interleukin IL1β*, *IL6*, *IL10*, and *tumor necrosis factor TNFα* was analyzed by RTD-PCR at day 0 and day 2, and TNFα secretion was analyzed by ELISA. The cytokine release in IPFPs was assessed by an array. Results: Both IPFPs (CO, OA) significantly reduced the expression of *type 2 collagen* and *TNFα* in chondrocytes. On the other hand, only CO-IPFP suppressed the expression of *type 1 collagen* and significantly induced the *MMP13* expression. On the contrary, *IL1β* and *IL6* were significantly induced when exposed to OA-IPFP. Conclusions: The partial loss of the suppressive effect on *type 1 collagen* gene expression found for OA-IPFP shows the pathological remodeling and dedifferentiation potential of the OA-IPFP on the chondrocytes. However, the significant suppression of *TNFα* implies that the OA- and CO-IPFP could also exhibit a protective role in the knee joint, preventing the progress of inflammation.

## 1. Introduction

Osteoarthritis (OA) is one of the most common joint diseases. Due to the limited capacity for intrinsic repair of mature cartilage, OA remains an orthopedic challenge. It is a disease that has a high impact on the single patient and is also a vast financial burden for the healthcare system [1]. Due to progressive, irreversible structural damage of the hyaline articular cartilage, OA leads to chronic severe pain and reduction in joint mobility [2]. According to the German Federal Office of Statistics, the diagnosis that was most frequently made in the whole outpatient sector was OA of the knee [3]. Approximately 25% of people aged over 55 develop knee pain. OA patients often report a significant decrease in their quality of life and productivity or even loss of independence [2,4]. Conservative symptomatic treatments in knee OA consist of an interplay between physiotherapy and painkillers. Injections of cortisone or hyaluronic acid in patients with mild to moderate OA are quite common, yet according to the literature, the effect is not significantly different from a placebo, and it cannot stop the progression of the disease [5]. The probability of developing OA increases with age, and in many cases, the patient’s therapy ends with a total joint replacement. Approximately 400,000 patients undergo total knee or hip arthroplasty every year in Germany alone [6]. Unfortunately, postoperative complications such as infection and loosening remain a high risk for the patients. The mean time to failure of total knee arthroplasty after the primary operation is, according to some authors, 5.9 years [7].

The etiology of OA is multifactorial [8]. It involves almost every structure of the knee and is currently accepted as a “whole joint disease” [9]. The progression of cartilage degeneration is not only caused by increased weight bearing but also by inflammation [10]. Under these circumstances, chondrocytes produce inflammatory mediators and degradative enzymes, such as matrix metalloproteinases (MMPs), which induce the self-degradation of joint cartilage. Metalloproteinase 3 was used as a biomarker in clinical studies, where it was shown that it correlates directly with the radiographic progression of OA in the knee [11]. All three MMPs in the knee joint used exemplarily in our study have a very high proteolytic potential, being possibly very important in the pathomechanism of OA.

One of the main players that has been taken into consideration within recent decades in the process of developing OA is the IPFP, also known as the Hoffa fat pad, first described by Albert Hoffa in 1904 [12]. It is a constant integral part of the knee joint and one of the fat bodies found in the anterior knee joint compartment aside from the supratrochlear and quadricep (suprapatellar) fat pads [13]. It was described as being significantly relevant for the mobility of the patella, patellar tendon, or tibia [14] and, hence, biomechanics. As an extra synovial part of the knee joint, it plays a key role in the pathogenesis of knee OA [13].

In vitro studies showed that the IPFP secretes pro- and anti-inflammatory cytokines and adipokines [15]. It was demonstrated that the IPFP from OA patients is fibrotic and inflamed, possessing a significantly higher gene expression and secretion profile of pro-inflammatory interleukin IL6 and its soluble IL6 receptor compared to the subcutaneous adipose tissue samples from the same donors [16]. Moreover, IPFP was also proven to secrete adipokines, e.g., leptin and adiponectin [17], both having pro-inflammatory properties on knee chondrocytes [18,19]. IPFP also possesses a number of substance-P-releasing nerve fibers [20] and peripheral nerve endings [21], which suggests a potential nociceptive role of the IPFP in the pathogenesis of knee pain syndrome. Furthermore, it was proposed that the IPFP represents a separate organ due to good innervation, vascularization, and its variety of cells, such as adipocytes, macrophages, lymphocytes, or even adipose-derived stem cells and mesenchymal stem cells (MSCs) [13].

The question of whether the IPFP plays a protective role or amplifies inflammation in the knee joint remains unknown. The aim of our study was to determine the direct influence of IPFP on cytokine expression in human chondrocytes to deepen the knowledge about its role in OA pathogenesis. To achieve this goal, we chose to study the expression of MMPs, collagen, and the expression and secretion of cytokines related to OA in human chondrocytes in a co-culture with a human IPFP. We compared the influence of IPFP ex vivo on cartilage degeneration based on the IPFP ex vivo from both patients struggling with OA and those without OA.

## 2. Materials and Methods

### 2.1. Sample Acquisition

Both articular cartilage (*n* = 11 donors) and the IPFPs (control IPFP *n* = 12 donors; OA-IPFP *n* = 9 donors) were obtained from knee replacement surgeries and arthroscopies performed in the Department of Traumatology and Reconstructive Surgery in the Charité—Universitätsmedizin Berlin, Campus Benjamin Franklin. Informed consent of all participating patients (approved by ethical regulation EA4/224/17) was obtained following the guidelines of the Charité Universitätsmedizin Berlin. Our donors were preselected based on the clinical examination and after a thorough assessment of the X-ray radiographs via Kellgren and Lawrence Scores. IPFP samples from patients with Kellgren–Lawrence grade 0, no clinical symptoms of knee abnormalities, and no knee pain were classified as control IPFP. Those patients had either meniscal lesions or cruciate ligament injuries. IPFP samples from patients having knee pain and/or motion abnormalities, as well as a Kellgren and Lawrence grade 4 score, were categorized as OA-IPFP samples [22,23]. The acquisition of the control IPFP samples was conducted during arthroscopies while viewing the central part of the fat pad using arthroscopy forceps. IPFP from OA patients were resected en bloc during total knee replacements. Because of the size difference between smaller control biopsies and larger OA-IPFP samples, we adjusted the samples (especially from the OA patients) by using the same-sized 6 mm stencil. Both control and OA-IPFP stencils were gathered from the same central part. Those adjusted samples were then used in further experiments. From the very beginning, patients with severe infections, immune defects, or undergoing any sort of immunomodulating therapy were excluded from the study. The age scale in the control collective measured 16–50-year-old patients (mean age in the control collective = 31.8 years old), and in the OA group, 50–81-year-old patients (mean age in the OA collective = 66.8 years old). The mean age of all participating donors was calculated at 53.6 years old.

### 2.2. Isolation of Human Articular Chondrocytes

All cartilage samples (*n* = 14) were obtained from 11 OA patients during the total knee replacement (approved by ethical regulation EA4/224/17). From the 11 OA patients included in the study, we gathered an OA cartilage sample from 6 patients, both an OA and a macroscopically intact control sample from 3 patients, and only a control sample from two patients. The cartilage specimens were obtained from the surgical ward and transported within 30 min in a phosphate-buffered solution supplemented with 1%-penicillin/streptomycin solution (Merck, Darmstadt, Germany) to the laboratory. First, every sample was macroscopically assessed as either OA- (osteoarthritis) or CO-type (control; healthy) based on the cartilage relief appearance. Samples from macroscopically detectable OA cartilage areas from weight-bearing regions, such as the tibial plateau, were categorized as OA cartilage (*n* = 9), and samples of macroscopically unchanged cartilage from non-weight-bearing regions, such as knee condyles and the intercondylar notch, were classified as the control cartilage (*n* = 5). Acquiring cartilage tissue from healthy patients would be against the regulations and ethical agreements. The human cartilage, cut into 1–3 mm small chips, was then enzymatically treated with 2% pronase E (Serva, Heidelberg, Germany) for 1 h and after that with 0.1% collagenase NB5 (Nordmark, Uetersen, Germany), both dissolved in Dulbecco’s modified Eagle’s medium (DMEM; Merck, Darmstadt, Germany). The digestion with collagenase took place on a Biosystem 4 Magnetic Stirrer (Thermo Fisher Scientific, Waltham, MA, USA) at 37 °C in a cell culture incubator (Thermo Fisher Scientific, Waltham, MA, USA) with 5% CO_2_ for 16 h overnight. Then, cells were filtered with a 70 µm cell strainer (Corning Incorporated, Corning, NY, USA) and washed twice, seeded in tissue culture flasks (Cell+; Sarstedt, Nümbrecht, Germany) at the density of 1 × 10^5^ cells/cm^2^, and expanded with expansion medium (DMEM/Ham’s F-12 1:1 medium with stable glutamine, supplemented with 10% fetal calf serum (FCS), 10,000 U/mL penicillin, 250 µg/mL Amphotericin B and 2.5 mg/mL ascorbic acid (all Merck, Darmstadt, Germany). After reaching confluence, cells were harvested and stored in 90% FCS + 10% DMSO (Merck, Darmstadt, Germany) at −80 °C until needed. Prior to their use in co-cultivation experiments, cell pools consisting of either 9 OA or 5 CO chondrocyte samples were prepared, and aliquots of these cell pools were cryoconserved for performing later independent co-culture experiments with the IPFP. In the result section, we pooled the results of the experiments independently conducted with the chondrocytes derived either from OA or CO samples (divided only in time lapses as d0 and d2). The chondrocyte experiments based on the biological origin (control vs. OA) were conducted separately throughout the whole study. The results achieved by independent experiments with chondrocytes of both sample types (OA and CO) were combined to increase the statistical power.

### 2.3. Co-Culture Experiments

On the day of the surgery, the IPFP samples were obtained from the patient and transported within 30 min to the laboratory in a phosphate-buffered saline (PBS; Merck, Darmstadt, Germany) supplemented with 1% penicillin/streptomycin. One pool of OA chondrocytes and one of the control chondrocytes derived from intact cartilage samples were defrosted, seeded in 6-well cell culture plates (Sarstedt, Nümbrecht, Germany), and cultivated in an expansion medium. IPFP punches were placed separately into well plates and covered with 2 mL 0.5%-FCS medium/well. After 2 days of incubation of IPFP tissues to adapt to culture conditions and to obtain a conditioned medium, IPFP samples were transferred together with their conditioned medium into inserts and put on the chondrocyte monolayer wells. The IPFP samples and the chondrocytes, initially 200,000 cells per PRO-6 well (20,000 cells/cm^2^), were separated with a 6-well insert with a 0.4 µm pore size membrane (Sarstedt, Nümbrecht, Germany) to create a co-culture milieu. The co-culture experiment was incubated for a further 2 days in a 0.5%-FCS medium at 37 °C with 5% CO_2_. The short time was selected considering the limited survival of immune cells in the IPFP. After that time, monolayer chondrocytes were lysed with RLT buffer (Qiagen, Hilden, Germany) supplemented with 1% β-mercaptoethanol (Merck, Darmstadt, Germany) for RNA extraction. Cell culture supernatants of IPFPs, chondrocyte cultures, and co-cultures were collected after 2 days of cultivation and stored at −80 °C.

### 2.4. RNA Extraction and RTD-PCR (Real-Time Detection Polymerase Chain Reaction)

The RNA extraction was performed using the RNeasy Mini Kit (Qiagen, Hilden, Germany), and the cDNA synthesis was conducted with the QuantiTect Reverse Transcription Kit (Qiagen, Hilden, Germany), both according to the manufacturer’s guidelines. The PCR was performed with a StepOnePlus cycler (Thermo Fisher Scientific, Applied Biosystems, Waltham, MA, USA) using SYBR Green PCR Master Mix (ThermoFisher Scientific, Waltham, MA, USA) as the master mix and GAPDH as the housekeeping gene. Thermal cycling parameters were divided into 2 chapters. In the first one, for 1 cycle, the temperature was set at 95 °C for 10 min for the time of the polymerase activation and then for another 36 cycles at the same temperature of 95 °C for 15 s (Denature) + 60 °C for 60 s (Anneal/Extend). The reaction volumes were set at 10 µL, template concentration at 1 ng/µL (10 ng per reaction), and primer concentration at 400 nM/400 nM (forward/reverse primer) as the final concentration.

All primers (Thermo Fisher Scientific, Waltham, MA, USA) used for gene expression analysis are listed in Table 1. Every single RTD-PCR experiment was performed as a triplicate of each sample. The *n*-values listed in the graphs used in the RTD-PCR do not represent the biological samples (cartilage donor numbers) but the number of independently conducted experiments performed with the CO and OA chondrocyte cell pool.

### 2.5. Histological Analysis by Hematoxylin and Eosin Staining (HE)

For histological analysis, approximately 1 cm^3^ IPFP sample was required. After washing, the tissues were transferred into centrifugation tubes filled with 4% paraformaldehyde (PFA; Santa Cruz Biotechnology, Dallas, TX, USA) for incubation for 1–2 days at 4 °C. Next, the samples were dehydrated for 24 h in 70% ethanol (Carl Roth, Karlsruhe, Germany) and for 5 h with an increasing ethanol concentration (from 80–99.6%), followed by clearing with xylol (Carl Roth, Karlsruhe, Germany) for 1 h. After dehydration, the samples were embedded in paraffin wax (Merck, Darmstadt, Germany) for 6 h at 60 °C. Slices up to 5 µm thickness were cut from the paraffin blocks with a rotary microtome HM 325 (Epredia, Braunschweig, Germany) and transferred to slides. After a dewaxing and rehydration process with xylol and ethanol (99.6–80%) for 20 min, all samples were treated with hematoxylin (Carl Roth, Karlsruhe, Germany) for 6 min. Then all samples were rinsed in distilled water, followed by 96% ethanol incubation for a further 2 min on the slides. At the end, all samples were stained with eosin (Sigma Aldrich, Steinheim, Germany) for 1.5 min, followed by the last dehydration step with ethanol (96–99.6%) for 8 min and xylol (Carl Roth, Karlsruhe, Germany) for 4 min before being covered with Entellan (Merck, Darmstadt, Germany).

### 2.6. ELISA

The TNFα secretion was measured in the supernatants of IPFP, chondrocyte cultures, and co-cultures from the co-culture experiments after 2 days. All the measurements were conducted using the Tecan Spectra Fluor Plus Microplate Reader (Tecan Trading AG, Männedorf, Switzerland, CH) and the BD OptEIA Human TNF ELISA Set (BD, Biosciences, Franklin Lake, NJ, USA). In contrast to the protocol guidelines, we used 0.5%-FCS medium for both the blanks and the standards. All of the experiments performed on the chondrocytes, IPFP, and co-cultures were conducted under the exact same circumstances. The supernatants from the chondrocyte and IPFP monocultures, as well as the co-cultures, were then compared with each other. The *n*-values listed in the graphs used in the ELISA do not represent the biological samples (cartilage donor numbers), but instead, the number of independently conducted experiments performed with the CO and OA chondrocyte cell pool.

### 2.7. Protein Array

To measure the amount of IPFP-secreted proteins, RayBio Cytokine Antibody Arrays (AAH-Cyt-1000; Raybiotech, Peachtree Corners, GA, USA) were performed for IPFPs from OA patients and IPFPs from healthy donors (*n* = 4 in each group). Total protein was isolated from 60 mg of IPFP tissue and processed for protein array measuring according to the manufacturer’s protocol. The protein suspension was diluted 1:5. Protein arrays were performed according to the manufacturer’s protocol. Chemiluminescence was detected using autoradiography films (Amersham Hyperfilm ECL; Cytiva, Marlborough, MA, USA) with an exposure time of 3–5 s. Films were scanned, and the chemiluminescence intensity of each sample was obtained by AlphaDigiDoc 1201 software (Alpha Innotech, Santa Clara, CA, USA) and correlated with the intensity of the antibody array’s internal positive control.

### 2.8. Statistical Analysis

Statistical analysis was conducted using the ROUT outlier test and one-way ANOVA test with Holm-Sidak’s multiple comparisons using GraphPad Prism version 10.0.0 and 6.07 for Windows (GraphPad Software, Boston, MA, USA, licence number: LIC29437, https://www.graphpad.com/). The *p*-value represented the level of significance of the data, with *p* ≤ 0.05 (*) considered significant, *p* ≤ 0.01 ** as very significant, *p* ≤ 0.001 *** as highly significant, and *p* < 0.0001 (****) as most significant.

## 3. Results

### 3.1. Histological Characteristics of the Control and OA Infrapatellar Fat Pads

After manual preparation, we investigated IPFP samples macroscopically. The OA-IPFP samples appeared considerably stronger vascularized than the control samples (Figure 1A,B). The control IPFP samples were collected from healthy individuals during arthroscopies and extracted in situ as 6 mm big biopsies.

Slides for histological staining were prepared and stained with hematoxylin and eosin (HE). The compact, cell-rich architecture of fatty tissue revealed a hexagonal configuration of the adipocytes shown in the HE stain. Between groups of adipocytes, small strands of dense collagenous connective tissue could be visualized. In contrast to the controls (Figure 2A), the OA-IPFP (Figure 2B) appeared more vascularized and contained more and thicker strands of the dense fibrous tissue.

### 3.2. Chondrocyte Gene Expression Profiles by Real-Time Detection Polymerase Chain Reaction (RTD-PCR)

In our study, we concentrated on the gene expression of three main *matrix metalloproteinases* (*MMP1*, *MMP3*, and *MMP13*), the *collagens type 1* (*COL1A1*) and *type 2* (*COL2A1*), the pro-inflammatory interleukins *IL1β* and *IL6*, the anti-inflammatory *IL10*, and of the *tumor necrosis factor* (*TNF)α* found in human cartilage extracellular matrix (ECM). Considering MMPs, only IPFPs from OA patients induced the gene expression of *MMP1* significantly (*p* = 1.3 × 10^−2^) in the co-cultivated chondrocytes (2.194 × 10^−1^ ± 4.886 × 10^−2^) compared to the basal expression in 2-day-cultivated chondrocytes without the IPFP (5.249 × 10^−2^ ± 8.811 × 10^−3^) (Figure 3A). On the contrary, no significant increase in gene expression of *MMP3* was observed in the co-cultivated chondrocytes—neither in the presence of control nor OA-IPFPs (Figure 3B). The gene expression of *MMP13* was significantly enhanced (*p* = 1.45 × 10^−2^), but only in the chondrocytes co-cultivated with the IPFP from healthy patients (7.157 × 10^−3^ ± 1.424 × 10^−3^) compared to the basal expression in 2-day-cultivated chondrocytes without the IPFP (2.953 × 10^−3^ ± 2.224 × 10^−4^) (Figure 3C). In fact, for all MMPs studied, the effects of control and OA-IPFP showed a tendency in the same direction without significant differences between them.

Collagens represent one of the main elements of the cartilage ECM. In our study, the gene expression of *COL1A1* in the co-cultivated chondrocytes was effectively (*p* = 1 × 10^−4^) suppressed in the co-culture with control IPFP from healthy patients after 2 days (1.344 × 10^−1^ ± 2.81 × 10^−2^) compared to chondrocytes cultivated in the absence of IPFP (4.59 × 10^−1^ ± 1.342 × 10^−3^). The downregulation of *COL1A1* gene expression in these chondrocyte co-cultures with control IPFP is significantly higher than the one caused by OA–IPFP (3.775 × 10^−1^ ± 6.272 × 10^−2^) (*p* = 1.2 × 10^−3^) (Figure 4A). Further, the gene expression of *COL2A1* was significantly inhibited by both control (1.356 × 10^−2^ ± 2.655 × 10^−3^) (*p* = 1 × 10^−3^) and OA-IPFP (1.539 × 10^−2^ ± 3.049 × 10^−3^) (*p* = 1.6 × 10^−3^) in the co-cultivated chondrocytes compared to the non-co-cultivated cells (3.445 × 10^−2^ ± 3.824 × 10^−3^) (Figure 4B).

In order to analyze the influence of OA-IPFPs on chondrocytes, we first took pro-inflammatory cytokines *IL1β* and *IL6* into consideration. Further, the gene expression of the anti-inflammatory cytokine *IL10* was also investigated in the IPFP/chondrocytes co-culture system. The chondrocytes co-cultivated with the OA-IPFP for 2 days showed significantly (*p* = 6 × 10^−4^ for *IL1β* and *p* = 1.92 × 10^−2^ for *IL6*) higher *IL1β* (6.853 × 10^−6^ ± 1.118 × 10^−6^) and *IL6* (2.105 × 10^−1^ ± 7.713 × 10^−2^) gene expression level compared to chondrocytes co-cultivated with CO-IPFP (2.136 × 10^−6^ ± 3.744 × 10^−7^ for *IL1β* and 2.054 × 10^−2^ ± 7.697 × 10^−3^ for *IL6*) and non-co-cultivated cells for 2 days (2.434 × 10^−6^ ± 5.086 × 10^−7^; *p* = 8 × 10^−4^ for *IL1β* and 1.005 × 10^−2^ ± 1.698 × 10^−3^; *p* = 8.4 × 10^−3^ for *IL6*) (Figure 5A,B). The gene expression of *IL10* was not changed in any of the analyzed groups (Figure 5C). However, the *TNFα* gene expression was significantly downregulated in the chondrocytes co-cultivated with CO-IPFP (4.979 × 10^−7^ ± 1.225 × 10^−7^; *p* = 7.8 × 10^−3^) as well as the OA–IPFP (5.713 × 10^−7^ ± 1.363 × 10^−7^; *p* = 9.5 × 10^−3^) compared to the non-co-cultivated chondrocytes at day 2 (1.48 × 10^−6^ ± 2.654 × 10^−7^) (Figure 5D).

### 3.3. TNFα Secretion

In order to determine the secreted levels of the pro-inflammatory cytokine TNFα, ELISA measurements were performed on the cell culture supernatants. The secretion of OA IPFP and IPFP from healthy patients was measured (*n* = 4). In addition, the secretion of cultivated chondrocytes (*n* = 14) was assessed. Co-cultivation of chondrocytes with OA and healthy IPFP (*n* = 8) was performed to see if a suppressive effect of TNFα secretion, as shown on the gene expression level, is feasible. All supernatants were collected after 2 days of cultivation. The IPFP itself showed a significantly lower TNFα secretion profile compared to the chondrocytes (hFP(OA) vs. hHC with *p* ≤ 1 × 10^−4^ and hFP(CO) vs. hHC with *p* ≤ 1 × 10^−4^). TNFα secretion in the co-cultivation assay was significantly lower than in cell culture supernatants of only the chondrocytes (hHC vs. hHC + hFP(CO) with *p* ≤ 1 × 10^−4^ and hHC vs. hHC + hFP(OA) with *p* ≤ 1 × 10^−4^) (Figure 6).

### 3.4. Protein Expression in IPFP

To measure the levels of IPFP secreted proteins in the IPFP tissue, a cytokine antibody array was performed for IPFPs from OA patients (*n* = 4) and IPFPs from healthy donors (*n* = 4). For leptin (Figure 7A), the mean normalized chemiluminescence signal (NCS) was slightly lower in IPFP samples from healthy donors (3.85 × 10^^−1^^ ± 1.27 × 10^−1^) compared to those of OA donors (6.69 × 10^−1^ ± 2.22 × 10^−1^). For adiponectin (Figure 7B), the level of NCS was higher (2.265 ± 4.51 × 10^−1^ for controls and 2.973 ± 1.37 × 10^−1^ for OA samples; *p* = 3.1409 × 10^−4^) than for leptin measured with the array. The mean value of adiponectin was 10× higher than the mean value of all other cytokines measured. XCL1, also known as lymphotactin (Figure 7C), showed a significantly higher NCS (*p* = 3.48469 × 10^−2^) in the control IPFP (4.64 × 10^−1^ ± 3.25 × 10^−1^) than IPFPs from OA patients (5.1 × 10^−2^ ± 4.1 × 10^−2^). IFN gamma (Figure 7D) showed an elevated level of the NCS in the IPFP from OA patients (9.9 × 10^−2^ ± 4.6 × 10^−2^) compared to the IPFP samples of the controls (2.2 × 10^−2^ ± 2.2 × 10^−2^).

Platelet-derived growth factor (PDGF-BB) (Figure 7E) showed an increasing trend in OA IPFP (1.7 × 10^−1^ ± 7 × 10^−2^ for controls and 4.48 × 10^−1^ ± 2.35 × 10^−1^ for OA samples). The results from the other proteins examined with the array can be found listed in detail in Appendix A.

## 4. Discussion

The aim of the study was to gain more insight into the interactions of IPFP with chondrocytes in OA pathogenesis. In particular, the role of IPFP in changes in gene expression of chondrocytes considering prominent inflammatory cytokines, MMPs, and collagens was targeted in this study. Whether IPFP has a positive or negative role in the knee joint still remains unknown. Therefore, we wanted to determine the influence of IPFP on human articular chondrocytes in an indirect in vitro co-culture assay to obtain deeper knowledge about its role in the OA pathogenesis.

Previous studies underlined an influence of IPFP-derived MSC on chondrocytes. IPFP-MSC in direct cell/cell co-cultures with chondrocytes revealed a supra-additive deposition of sulfated glycosaminoglycans (sGAG), collagen type 2, aggrecan, and link protein [24]. Furthermore, extracts from OA-chondrocytes induced IPFP-derived cells to produce cartilage-like ECM [25]. These results demonstrate that OA-chondrocytes can influence IPFP cells. Nevertheless, no information about changes in cytokine levels is known. Our approach for an indirect co-culture was the exact opposite of the mentioned co-culture examples. In the chosen assay, there is no direct cell-tissue contact. Due to cell culture plate inserts, the IPFP is separated by a membrane from the monolayer chondrocytes. This shall mimic the synovial membrane separating the IPFP from the knee joint cavity. Due to the high permeability of the synovial membrane, a direct influence of the mediators of the Hoffa fat pad on the articular cartilage, especially in an inflammatory phase of the knee joint, is possible.

For MMPs, the co-cultures only showed slight changes. All measured MMPs demonstrated an elevated mean normalized gene expression in chondrocytes co-cultured with IPFP compared to those without co-culture. However, only co-culture with OA-derived IPFP had a significant inductive influence on MMP1 and co-culture with IPFP from healthy donors on MMP13 gene expression. In general, the different MMP expressions in co-cultures imply changes in the structure and content of ECM. In former studies, MMP expression led to different results [26]. On the one hand, conditioned medium from OA knee joint IPFP induced cartilage collagen release and increased MMP1 and MMP13 expression in bovine chondrocytes [27]. On the other hand, IPFP-conditioned medium from tissues of OA patients reduced the gene expression of MMP1 in bovine cartilage [28]. Nevertheless, in our study, unlike in many other studies, we used human IPFP samples from OA and healthy patients ex vivo and also chondrocytes from human patients, revealing slightly different results compared to the bovine chondrocytes in other studies.

Collagens are the main components in the ECM of cartilage. The IPFP is considered to be a source of MSCs with a high differentiation potential towards the chondrogenic lineage [24]. Interestingly, COL1A1, being associated with the dedifferentiation of chondrocytes, was significantly decreased in co-cultures with fat pads of healthy donors but not in those with OA fat pads and the chondrocyte monolayer culture without fat pad co-cultivation. COL2A1 expression of chondrocytes was significantly downregulated in both types of co-cultures. The significant downregulation of the gene expression of collagens in the chondrocytes co-cultivated with the control IPFP emphasizes a potential role of IPFP-derived mediators in the cartilage remodeling processes. The downregulation of COL1A1 in co-cultures with healthy IPFP could indicate a role of healthy IPFP in suppressing pathological remodeling and dedifferentiation processes. In former studies, conditioned media from OA knee joint IPFP induced cartilage collagen release in human articular chondrocytes [29] and also bovine cartilage based on an animal experiment [27]. Our study is one of the first ones to investigate the difference in the cytokine levels of IPFP from OA and healthy human donors ex vivo.

Cytokines play an important role in inflammatory processes. In vitro studies demonstrated that the Hoffa fat body secretes pro- and anti-inflammatory cytokines and adipokines [30] and, therefore, influences the expression of cytokines in chondrocytes. In our study, we could see a highly significant increase in *IL1β* and *IL6* gene expression in chondrocytes co-cultured with OA fat pad. This was not visible in co-cultures with IPFP from healthy donors. Immune cells in IPFP, such as neutrophils, eosinophils, and basophils, are shown to produce degrading enzymes and proinflammatory mediators, such as interleukins and metalloproteinases, while the adipocyte cells are responsible for the production of adipokines, such as leptin, adiponectin, and resistin [13,31]. Leptin is known to increase IL6 levels and induce the expression of MMPs in the OA cartilage [32]. Its content in IPFP did not significantly differ between OA patients and controls, with only a trend of higher expression in OA-IPFP. Leptin could be connected to OA via its capacity to upregulate LOXL3. LOXL3 was found to be overexpressed in OA-affected cartilage, and after leptin exposure, the LOXL3 levels were upregulated. Elevated levels of LOXL3 induced higher levels of pro-apoptotic factors, such as cleaved caspase 3, exerting catabolic effects on primary chondrocytes [33]. Another protein that indicated inflammation was the IFN gamma. The catabolic effect of IFN gamma on cartilage tissue could result from the activation of the gene expression of the transcription factor STAT1 [34]. Similarly, it was shown that the platelet-derived growth factor (PDGF-BB) could be connected to OA via its role in subchondral bone angiogenesis [35]: After destabilization of the medial meniscus in mice knees, mononuclear preosteoclasts start to overexpress the PDGF-BB that signals the expression of the platelet-derived growth factor receptor-ß. This leads to activation of the subchondral bone angiogenesis, which is known as an early stage of OA [35]. However, PDGF-BB content in IPFP did not significantly differ between OA patients and controls.

In IPFP cells, adiponectin induces MMP1 and IL6 production of synovial fibroblasts [36]. In agreement with this observation, chondrocytes treated with adiponectin produce a higher amount of IL6 [37]. Another study reported that the proinflammatory function of the full-length adiponectin was derived from the significantly increased MMP-13 activity and PGE2 synthesis, contributing to the catabolic state of the cartilage [38]. Adiponectin is also known to possess anti-inflammatory properties. When secreted from adipose tissue, it inhibits the effect of TNFα and directly counter-regulates TNFα production [39]. In our case, TNFα gene expression in chondrocytes in co-culture with IPFPs from OA and healthy tissues was downregulated compared to the control monolayer culture without IPFP co-culture. Both IPFP from healthy as well as OA patients led to significantly decreased gene and protein expression. During the 2 days of cultivation, a significantly higher amount of TNFα was secreted by the chondrocytes without co-cultivation with any IPFP. The secretion of TNFα by IPFPs alone or by the IPFP co-cultivated with chondrocytes was on a similar significantly lower secretion level than in chondrocyte cultures. Additionally, we also found a high secretion of adiponectin in both IPFP groups (OA and healthy) in the protein array. Therefore, as mentioned above, the adiponectin released by the IPFP independently from the OA grade might be the reason for the reduced TNFα secretion and gene expression. Other adipokines might also be involved in cytokine regulation during the co-culture. Resistin, for example, is one of the adipokines that stimulates the production of IL1β, IL6, and TNFα in peripheral blood mononuclear cells (PBMCs) and, therefore, contributes directly to inflammation [32,40]. Another protein investigated in the present study was lymphotactin. Compared to leptin and adiponectin, lymphotactin showed clear anti-inflammatory features. The chondroprotective role of this particular chemokine might come from its ability to recruit human subchondral mesenchymal progenitor cells significantly and induce anabolic features by cartilage renewal [41].

Although we know by now that the IPFP has a certain impact on the formation of OA, there is still a range of factors that have to be considered in the pathophysiology of OA and the influence of the IPFP.

One major role is the overall body constitution of a test person and their health status in general, as it was stated that, e.g., obesity and metabolic syndrome are some of the main risk factors for OA [42,43]. It was proven that IPFP adipocytes modulate cytokine secretion from CD4+ T cells. This means that patients with severe immune defects, undergoing immunotherapy, or having multiple banal transient infections might show a different secretion of some cytokines, such as TNFα, IL17, and IL5, caused by a generally suppressed immune system [44]. In vivo, the IPFP volume appeared greater in OA samples than in the controls. This phenomenon was described in a certain study showing that the fat pad size clearly depends on the OA itself and does not correlate with the age of the patients that the samples are acquired from [45]. This is an important observation in view of the differing mean ages of the IPFP human donor cohorts in the present study. Unfortunately, it is difficult to obtain healthy IPFP from older patients since indications for arthroscopy are more often given in younger persons. This is a clear limitation of the present study in working with human samples.

During arthroscopies, we could cut out only a very small representative piece of the IPFP from control samples, whereas IPFP from a total knee replacement was basically a side product of the whole intervention itself. After gathering the IPFP samples, in order to make a comparison between those two, OA tissue was cut with a stencil (6 mm diameter) to make the size of it comparable with the CO-tissue. There was no vitality analysis performed prior to our IPFP co-culture experiments as there was a proven cell vitality protocol based on a previous study conducted on tenocytes and immune cells with the exact same cell culture medium [46]. Similarly, as the arthroscopy procedure does not require the removal of cartilage per se, chondrocyte samples used in our experiments were only gathered from OA patients. The removal of control cartilage during arthroscopies without clinical indication from patients, independent of their age, would be strictly against the ethical regulations of our study.

OA is, in fact, a polymorphous and multifactorial disease, where the genetics of the individual might also play a crucial role in the development of the OA [47,48], modulating the intensity of other risk factors. Genetic variations were considered an exclusion factor. Recent findings suggest that the fat pad with its synovium should be seen as one “anatomo-functional unit” [49]. In our study, we only examined the influence of the fat pads from OA and healthy donors on chondrocytes in vitro by using samples from the center of every IPFP. This might give us a hint of the in vivo processes without mirroring the exact conditions of the interactions occurring in the complex molecular environment in vivo. In addition to the effect on chondrocytes, IPFP most probably plays a significant role in the development of OA by secreting pro-inflammatory cytokines into the synovial fluid [50].

Taken together, IPFPs from both groups reduced the gene expression of COL2A1 and TNFα. Results of TNFα were confirmed by protein level. In addition to these similarities, we also noticed some differences concerning the influence of the control and OA fat pad on the gene expression of the examined cytokines. IPFPs from patients suffering from OA are clearly inducing, via soluble mediators, an increased gene expression of the inflammatory cytokines IL1β and IL6 in chondrocyte cultures. COL1A1, on the contrary, was significantly downregulated in co-cultures with tissues from healthy patients. Adipokines might play a certain role in the modulation of these processes. These results underline the potential influence of the IPFP in the pathogenesis of OA, considering inflammation and remodeling processes. However, OA is a disease that requires multiple simultaneous therapeutic approaches in different fields. Nevertheless, by trying to acquire a deeper insight into its molecular components and their interactions with each other, we are definitely moving in the right direction towards a deeper understanding of the complex pathology of the OA.

## Figures and Tables

**Figure 1 cells-12-02850-f001:**
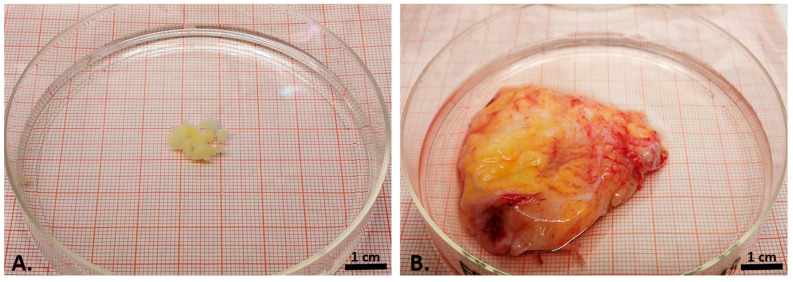
Macroscopical appearance of the biopsy from the control infrapatellar fat pad (**A**) and the explanted OA infrapatellar fat pad (**B**), both shown on the same-sized petri dish.

**Figure 2 cells-12-02850-f002:**
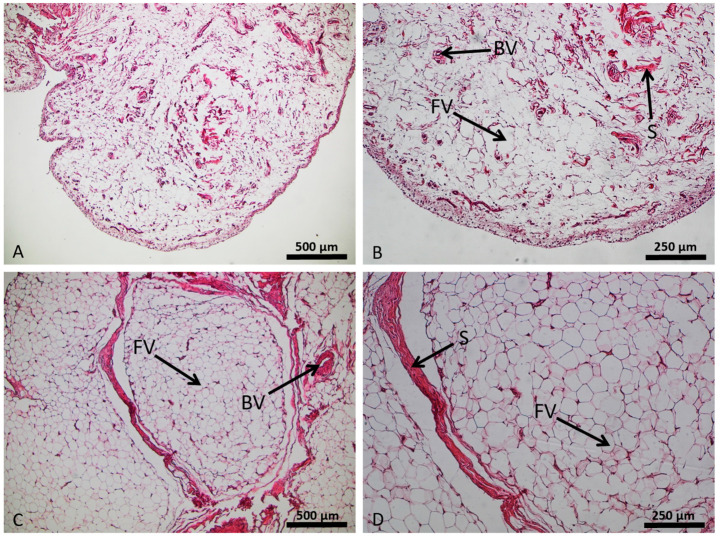
Structural setup of the adipose infrapatellar fat pad tissue (control sample in (**A**,**B**); osteoarthritis sample in (**C**,**D**)), hematoxylin and eosin staining. Lipid vacuoles of adipocytes (FV), blood vessels (BV), and stroma (S).

**Figure 3 cells-12-02850-f003:**
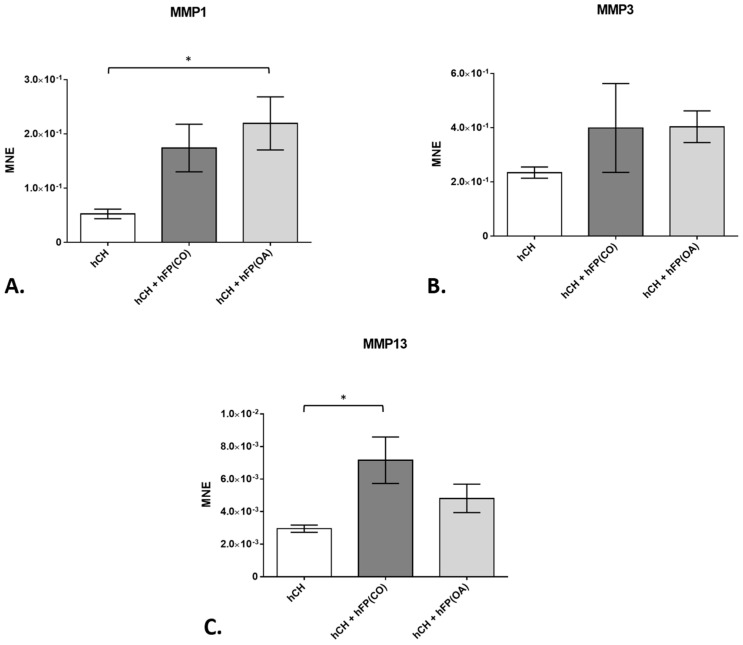
Gene expression of matrix metalloproteinases (*MMP*) in chondrocyte cultures exposed to or not exposed to human infrapatellar fat pad tissue (hFP); analyzed by one-way ANOVA using Holm-Sidak’s multiple comparison tests. The graph shows the gene expression as the mean normalized expression value (MNE) of *MMP1* (**A**), *MMP3* (**B**), and *MMP13* (**C**) detected in the following chondrocyte samples: hCH represents pre-cultivated human articular chondrocytes (hCH) cultivated for 2 days (*n* = 14), hCH + hFP(OA) represents chondrocytes co-cultivated with OA IPFP for 2 days (*n* = 10), and hCH + FP(CO) represents chondrocyte samples co-cultivated with control IPFP for 2 days (*n* = 12). *p* ≤ 0.05 *. Data are presented as mean +/− standard error of the mean (SEM).

**Figure 4 cells-12-02850-f004:**
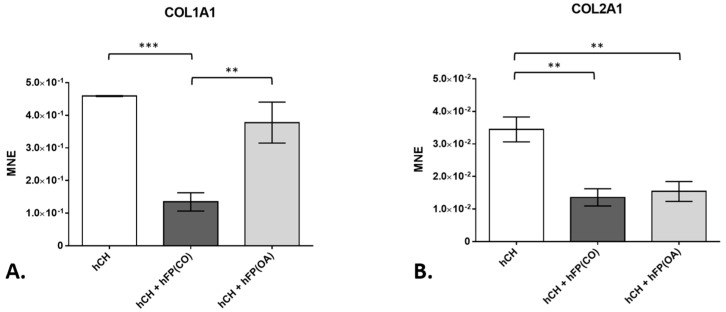
Gene expression of *collagen type 1* (*COL1A1*) and *collagen type 2* (*COL2A1*) in chondrocyte cultures exposed to or not exposed to infrapatellar fat pad tissue (hFP); analyzed by one-way ANOVA using Holm-Sidak’s multiple comparison tests. The graph shows the gene expression as the mean normalized expression value (MNE) of *COL1A1* (**A**) and *COL2A1* (**B**) detected in the following chondrocyte samples: hCH represents pre-cultivated human articular chondrocyte (hCH) samples cultivated for 2 days (*n* = 6), hCH + hFP(OA) represents co-cultivated chondrocytes with OA IPFP for 2 days (*n* = 6), and hCH + hFP(CO) represents chondrocytes co-cultivated with control IPFP for 2 days (*n* = 6). *p* ≤ 0.01 **, *p* ≤ 0.001 ***. Data are presented as mean +/− standard error of the mean (SEM).

**Figure 5 cells-12-02850-f005:**
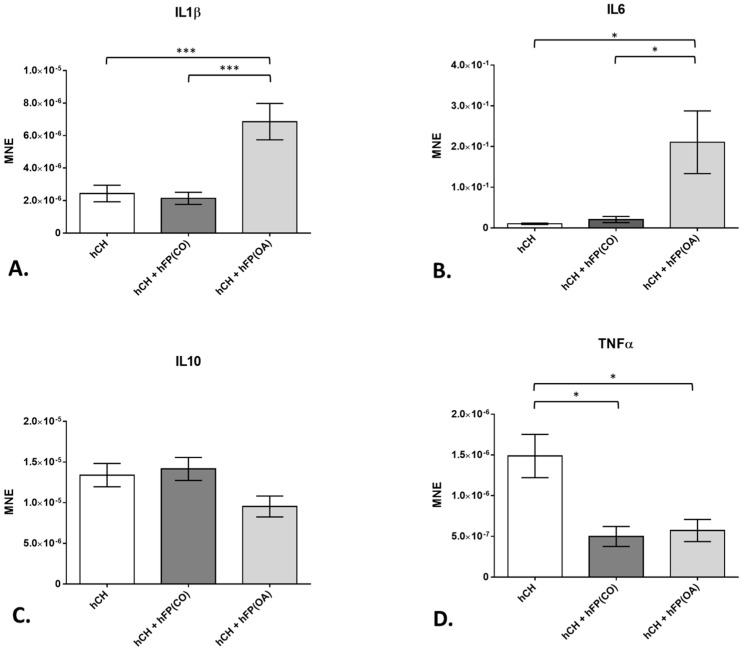
Gene expression of cytokines in the chondrocyte culture exposed to or not exposed to human infrapatellar fat pad tissue (hFP); analyzed by one-way ANOVA using Holm-Sidak’s multiple comparison tests. The graph shows the gene expression as the mean normalized expression value (MNE) of interleukins *IL1β* (**A**), *IL6* (**B**), *IL10* (**C**) and tumor necrosis factor α (**D**) detected in the following chondrocyte samples: hCH represents pre-cultivated human articular chondrocyte (hCH) cultivated for 2 days (*n* = 14), hCH + hFP(OA) represents chondrocytes co-cultivated with OA IPFP for 2 days (*n* = 10), and hCH + FP(CO) represents chondrocytes co-cultivated with control IPFP for 2 days (*n* = 12). *p* ≤ 0.05 *, *p* ≤ 0.001 ***. Data are presented as mean +/− standard error of the mean (SEM).

**Figure 6 cells-12-02850-f006:**
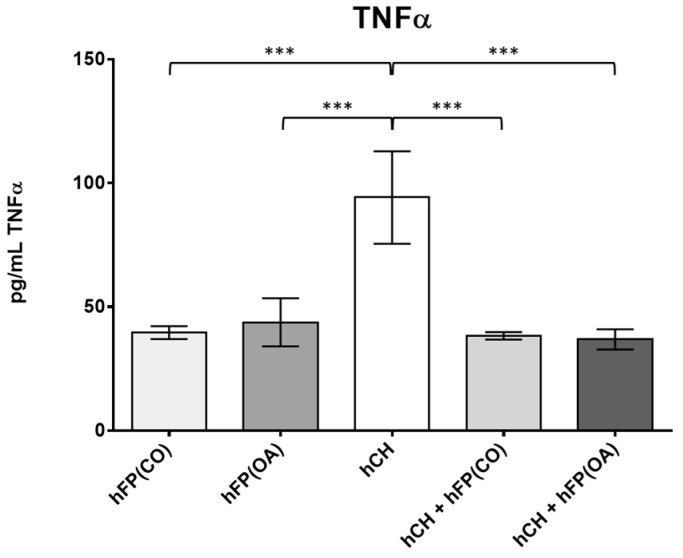
Secretion profile of TNFα (in pg/mL) in the measured supernatants analyzed by one-way ANOVA using Holm-Sidak’s multiple comparison tests. The graph shows the secretion level of TNFα (in pg/mL) detected in the following supernatant samples: hFP(CO) represents supernatants of control IPFP samples of healthy patients and hFP(OA) from OA samples cultivated for 2 days (*n* = 4), hHC represents supernatants from chondrocytes cultivated for 2 days (*n* = 16) and hCH + hFP(CO), and hCH + hFP(OA) represent supernatants of co-culture experiments (*n* = 8). *p* ≤ 0.001 ***. Data are presented as mean +/− standard error of the mean (SEM).

**Figure 7 cells-12-02850-f007:**
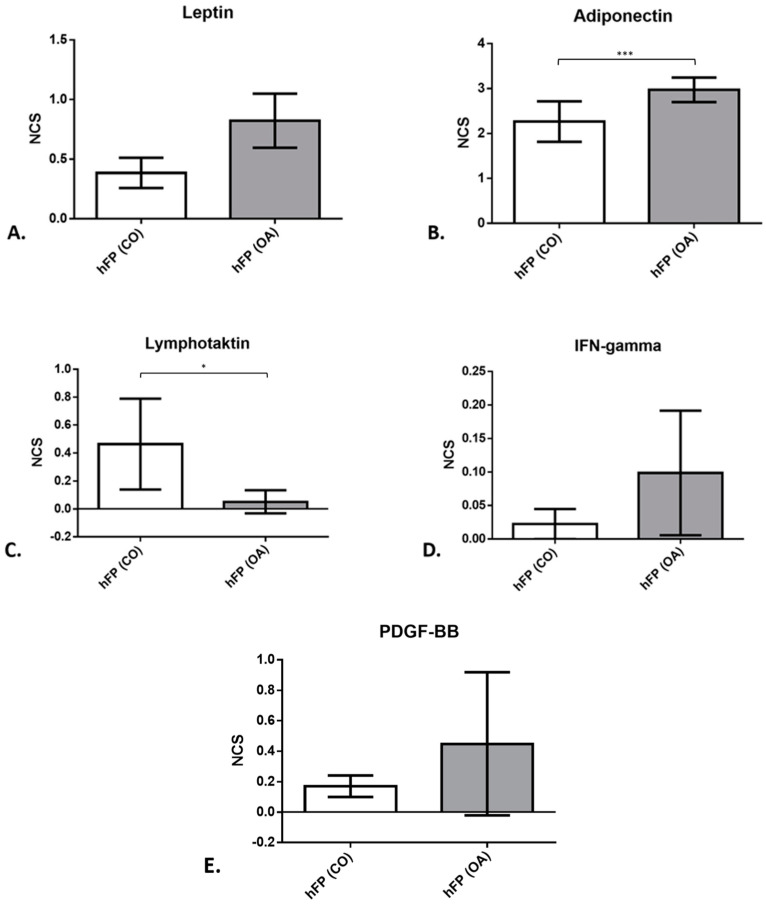
Protein expression (leptin (**A**), adiponectin (**B**), lymphotactin (**C**), IFN gamma (**D**), and PDGF-BB (**E**)) of IPFP tissue from control samples of healthy patients hFP(CO) and hFP(OA) from OA samples directly after explantation, given as the mean normalized chemiluminescence signal (NCS). *p* ≤ 0.05 *, *p* ≤ 0.001 ***. Data are presented as mean +/− standard error of the mean (SEM).

**Table 1 cells-12-02850-t001:** Gene names with the corresponding accession number (NM), the exact primer sequences, and the PCR product size measured in base pairs (bp) used in the RTD-PCR.

Gene	NM-Number	Forward/Reverse Primer Sequence	PCR Product Size (bp)
*MMP1*	NM_002421	5′ GGGAATAAGTACTGGGCTGTTCA 3′ 3′ TCCAGGAAAGTCATGTGCTATCA 5′	231
*MMP3*	NM_002422	5′ CTATCAGAGGAAATGAGGTACGAGC 3′ 3′ GCCTGGCTCCATGGAATTTC 5′	179
*MMP13*	NM_002427.3	5′ CCCCAGGCATCACCATTCAA 3′ 3′ CAGGTAGCGCTCTGCAAACT 5′	150
*IL1β*	NM_000576	5′ CCCTAAACAGATGAAGTGCTCC 3′ 3′ AGAAGGTGCTCAGGTCATTCTC 5′	197
*IL6*	NM_000600.4	5′ GGCACTGGCAGAAAACAACC 3′ 3′ GCAAGTCTCCTCATTGAATCC 5′	85
*IL10*	NM_000572.2	5′ AAGACCCAGACATCAAGGCG 3′ 3′ AATCGATGACAGCGCCGTAG 5′	85
*COL1A1*	NM_000088	5′ GGCAACGATGGTGCTAAGG 3′ 3′ GACCAGCATCACCTCTGTCA 5′	139
*COL2A1*	NM_001844	5′ GATGGCTGCACGAAACATACC 3′ 3′ AAGAAGCAGACCGGCCCTAT 5′	155
*TNFα*	NM_000594.3	5′ ATGTTGTAGCAAACCCTCAAGC 3′ 3′ CTTGGTCTGGTAGGAGACGG 5′	227
*GAPDH*	NM_002046	5′ GAAGGTGAAGGTCGGAGTC 3′ 3′ GAAGATGGTGATGGGATTTC 5′	226

Bp: base pairs.

## Data Availability

Due to the privacy of the patients involved, we are not allowed to release patient data.

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
