# Peer review of "Infrapatellar Fat Pad Modulates Osteoarthritis-Associated Cytokine and MMP Expression in Human Articular Chondrocytes"

_cells, 2023, doi:10.3390/cells12242850_

Round 1
Reviewer 1 Report (Previous Reviewer 2)
Comments and Suggestions for Authors
The reviewer thanks the authors for thoroughly answering and resolving the raised questions.
Author Response
Dear reviewer.
Thank you for prompt and positive response.
Sincerely,
M. Jagielski
Reviewer 2 Report (New Reviewer)
Comments and Suggestions for Authors
The study is interesting. I have several comments for the authors.
It must be clearly reported what the patients undergoing arthroscopic surgery (control IPFP) had.
Introduction on IPFP should be improved. It should be reported that OA IPFP is inflamed and fibrotic. There is also no mention about the biomechanical role of this tissue.
Lines 75-76: It has been recently shown that IPFP presents also peripheral nerve endings (positivity to PGP9.5) (doi:10.3389/fcell.2022.886604).
Line 97-99: What did these patients (control patients) have? It needs to be clarified.
Line 105: did the authors weigh the IPFP samples? Mean and SD of the IPFP of both groups should be added.
Lines 115-126: this part is unclear. In particular, it is unclear what chondrocytes did the authors use in this study. Did the authors use OA chondrocytes or “control” chondrocytes isolated from OA patients?
Lines 149-152: this is reported also at lines 102-105.
Lines 156: why after 2 days of incubation?
Line 157: how many chondrocytes did the authors seed?
Line 159: why 2 days? Are there other studies supporting this?
Lines 205-214: instrument used should be added.
Lines 231-232: Graphpad should be cited as reported in website of the software: https://www.graphpad.com/guides/prism/latest/user-guide/citing_graphpad_prism.htm
Lines 238-239: it should be clearly report what are control IPFP.
Lines 243-248: the authors simply described the histological images. It would be useful to evaluate infiltration, vascularization etc using IPFP score published (doi: 10.1093/rheumatology/kex287), in order to perform statistical analysis between the two groups.
Section 3.2: three numbers after the decimal point could be enough.
The authors should consider using the 2^ddct method to elaborate real time PCR, in order to avoid to obtain numbers with a lot of zero after the decimal point.
The authors reported in the methods that the human cytokine array C1000 of Raybiotech was used. Is this: https://www.raybiotech.com/human-cytokine-array-c1000-aah-cyt-1000 ? If yes, this kit enable the quantification of 120 cytokines but only 5 proteins were reported in the results.
Why did the authors select to evaluate only TNF-beta in the supernatant? At least IL1 beta secretion should be determined.
Lines 478-479: this part needs to be improved. It was not mentioned at all that it has been published a study demonstrating as there is an age-dependent remodeling of the infrapatellar fat pad.
Author Response
Please see the attachment

Reviewer 3 Report (New Reviewer)
Comments and Suggestions for Authors
The aim of this study was to gain more insight in the interactions of IPFP with chondrocytes in OA pathogenesis. For this the authors performed cocultures of OA chondrocytes in monolayer with 6 mm diameter punches taken from healthy or OA IPFP, in a transwell system. They measured gene expression in the chondrocytes, TNF-a protein in the culture supernatant, and proteins present in the IPFP tissue. Effects of heathy an OA IPFP were often comparable, but for some genes the effect of the OA IPFP was significantly different from the effect of the healthy IPFP on chondrocytes. To my opinion those are the most important, in the light of the aim of the study. The same holds for the significant decrease of lymphotaxin content of OA IPFP, causing loss of its protective capacity.
My comments:
Abstract, line 18: “IPFP harvested from healthy patients harvested during knee arthroscopy served as controls” This gives the impression that the whole IPFP was taken. The authors could better state that they took a biopsy from the IPFP in these patients.
Abstract, lines 23-30: The description of the results here is rather random. I miss the focus. I would first describe the effects that control and OA IPFP have in common and next where they significantly differ (effects on IL-1β, IL-6 and Col1A1 gene expression in chondrocytes), and the implications of this. In the conclusions the most important results and their implications should be stated, and now this is not the case. The authors mention there that the IPFP could play a crucial role in OA by differentially regulating cartilage remodeling through MMPs. This conclusion is wrong, because all MMPs studied were upregulated, and no significant differences were found between the effects of the control and OA IPFP here. The next conclusion is valid, but more in line with the aim of the study, Instead, I would highlight here the partial loss of suppressive effect on Col1A1 gene expression found for OA IPFP, and its possible implications.
Introduction, line 44: “new pharmacological approaches like disease-modifying OA drugs” Here the authors refer to a study published in 2016, “Disease-modifying treatments for osteoarthritis (DMOADs) of the knee and hip: lessons learned from failures and opportunities for the future”. To my knowledge there are still no DMOADs available up to the present. This means that they could not be involved in the current treatments.
Introduction, line 60: “All three” where do the authors refer to? There are more than 3 MMPs present in the joint.
Introduction, lines 67-68: “Some authors describe a strong correlation between the IPFP and severity of the OA” Here we miss the correlating parameters that were measured in the IPFP.
Paragraph 2.2, line 118: “control sample”. To be clear, the authors could add here “macroscopically intact/unaffected or normal looking” control sample.
Paragraph 2.5, line 202: “before covered” should be “before being covered”
Paragraph 3.1, linees237-239: this is confusing to me. This line gives me the impression that this re both complete IPFPs, but from the rest of the text I would expect that you have only permission to take a small biopsy from non-OA patients during arthroscopy. In that case it would be not very surprising that the control biopsy is much smaller than a complete OA IPFP. Please make this clear.
Paragraph 3.2, lines 256-268: This part could be finished by stating that in fact for all MMPs studied the effects of control and OA IPFP were in the same direction, without significant differences between the two.
Figures 3-7, legends: ”data are presented as standard error of the mean (SEM)” I would say “data are presented as mean +/- standard error of the mean (SEM)”.
Legends figure 5, line 314: “gene expression of cytokines in the chondrocytes and coculture samples”. This is confusing, because this gene expression was always measured in the chondrocytes. Better use the same construction of the sentence as used in figure 3, with “chondrocyte cultures exposed to or not exposed to human infrapatellar fat pad tissue”.
Paragraph 3.3, lines 333-334: “lower than in chondrocytes itself” I propose “lower than in cell culture supernatants of only the chondrocytes”.
Paragraph 3.4, line 346: Instead of “To measure the influence of IPFP secreted proteins”, I would say “To measure the levels of IPFP secreted proteins in the IPFP tissue”, to make things more clear.
Paragraph 3.4, line 358: “Platelet-derived growth factor showed similar like….catabolic features on the human cartilage” It is not clear how the underlined part is connected to the results presented in this manuscript. I propose to change this part of the sentence for “a trend to increase in OA IPFP”.
Discussion, lines 425-426: Leptin content in IPFP did not significantly differ between OA patients and controls, so the authors should mention this here.
Discussion, line 427: Besides the just mentioned absence of significance, this sentence is very strange. The authors write here that a possible pathogenetic reason for higher leptin in OA IPFP could be that leptin upregulates LOXL3. I think that the authors mean to say that leptin could be connected to OA via its capacity to upregulate LOXL3.
Discussion, line 433: PDGF-BB content in IPFP did not significantly differ between OA patients and controls, so the authors should mention this here. There is high variability between the effects of OA IPFPs of OA patients on PDGF-BB content.
Discussion, lines 436-440: Besides the just mentioned absence of significance, the next sentences are very strange. Elevated NSC levels of PDGF-BB in the IPFP OA samples would be explained by the findings in a mouse experiment. I think that the authors mean to say that PDGF-BB could be connected to OA via its role in subchondral bone angiogenesis.
Discussion, lines 454-455: “Therefore, the IPFP has suppressed the TNF-α production of the chondrocytes. This conclusion is not valid, because both the chondrocytes and the IPFP produce TNF-α, if cultured alone. When cultured together, the level is lower, but this could also mean that chondrocytes have suppressed TNF-α production of the IPFP.
Discussion, Lines 507-514: In this concluding part of the discussion, again I would first mention what effects control and OA IPFPs have in common , and end with the effect that are significantly different in OA IPFP, because these are most relevant considering the aim of the study. Then MMPs do not need that much attention any more.
Discussion, lines 515-516: “These results underline the potential influence of the IPFP on human cartilage its certain role in the pathogenesis of OA, considering inflammation and remodeling processes”. Besides the strange construction of this sentence, I also have the comment that the authors should stay close to their own findings. There was no proof for a certain role of the IPFP in the manuscript, so I propose to confine this sentence to “These results underline the potential influence of the IPFP in the pathogenesis of OA, considering inflammation and remodeling processes”.
Author Response
|
1. Summary |
|
|
|
Thank you very much for taking the time to review this manuscript. Please find the detailed responses below and the corresponding revisions/corrections in the re-submitted files.
|
||
|
2. Point-by-point response to Comments and Suggestions for Authors
Comments 1: Abstract, line 18: “IPFP harvested from healthy patients harvested during knee arthroscopy served as controls” This gives the impression that the whole IPFP was taken. The authors could better state that they took a biopsy from the IPFP in these patients. Response 1: It is a very important advice. We added the note that it was a biopsy (line 18).
Comment 2: Abstract, lines 23-30: The description of the results here is rather random. I miss the focus. I would first describe the effects that control and OA IPFP have in common and next where they significantly differ (effects on IL-1β, IL-6 and Col1A1 gene expression in chondrocytes), and the implications of this. In the conclusions the most important results and their implications should be stated, and now this is not the case. The authors mention there that the IPFP could play a crucial role in OA by differentially regulating cartilage remodeling through MMPs. This conclusion is wrong, because all MMPs studied were upregulated, and no significant differences were found between the effects of the control and OA IPFP here. The next conclusion is valid, but more in line with the aim of the study, Instead, I would highlight here the partial loss of suppressive effect on Col1A1 gene expression found for OA IPFP, and its possible implications. Response 2: Thank you for pointing that out and helping us shift our focus on the relevant information in the study. We changed the order and categorized the findings as it was suggested in the revision (lines 23- 30).
Comment 3: Introduction, line 44: “new pharmacological approaches like disease-modifying OA drugs” Here the authors refer to a study published in 2016, “Disease-modifying treatments for osteoarthritis (DMOADs) of the knee and hip: lessons learned from failures and opportunities for the future”. To my knowledge there are still no DMOADs available up to the present. This means that they could not be involved in the current treatments. Response 3: For the sake of precision we deleted this potential treatment option from the text and left only the established therapeutic options like analgesia and physiotherapy (lines 42- 44)
Comment 4: Introduction, line 60: “All three” where do the authors refer to? There are more than 3 MMPs present in the joint. Response 4: That is a very good and precise point of critic. We meant all 3 MMPs from the knee joint that we used as an example in our study, meaning MMP1, MMP3 and MMP13. It was a kind of a brachylogy. We added the necessary information to the sentence (line 60).
Comment 5: Introduction, lines 67-68: “Some authors describe a strong correlation between the IPFP and severity of the OA” Here we miss the correlating parameters that were measured in the IPFP. Response 5: After re- reading the thesis, we decided to leave that sentence out at this place, as the study cited here does not reasonably support this sentence, by missing a clear parametrial correlation. Moreover the statement does not fully fit in the rest of the written context here.
Comment 6: Paragraph 2.2, line 118: “control sample”. To be clear, the authors could add here “macroscopically intact/unaffected or normal looking” control sample. Response 6: We implicated the suggestion by adding the phrase ´´macroscopically intact´´ (line 120).
Comment 7: Paragraph 2.5, line 202: “before covered” should be “before being covered” Response 7: We changed the phrase as suggested above (lines 208- 209).
Comment 8: Paragraph 3.1, lines 237-239: this is confusing to me. This line gives me the impression that this are both complete IPFPs, but from the rest of the text I would expect that you have only permission to take a small biopsy from non-OA patients during arthroscopy. In that case it would be not very surprising that the control biopsy is much smaller than a complete OA IPFP. Please make this clear. Response 8: This picture shows on the right side the whole IPFP from an OA donor gathered after a total knee arthroplasty (being basically a side product of the intervention) and on the left side a small biopsy (6mm stencil) from a healthy donor, gathered during a arthroscopy (Figure 1). It is more clearly stated in the legend of figure 1 now (lines 250- 251).
Comment 9: Paragraph 3.2, lines 256-268: This part could be finished by stating that in fact for all MMPs studied the effects of control and OA IPFP were in the same direction, without significant differences between the two. Response 9: The finishing conclusion was added as suggested (lines 277- 279).
Comment 10: Figures 3-7, legends: ”data are presented as standard error of the mean (SEM)” I would say “data are presented as mean +/- standard error of the mean (SEM)”. Response 10: The sentence got changed according to the suggestion (Figures 3-7).
Comment 11: Legends figure 5, line 314: “gene expression of cytokines in the chondrocytes and coculture samples”. This is confusing, because this gene expression was always measured in the chondrocytes. Better use the same construction of the sentence as used in figure 3, with “chondrocyte cultures exposed to or not exposed to human infrapatellar fat pad tissue”. Response 11: The description got clarified according to the suggestion in Fig. 4 and 5 (lines 303- 304 and 328- 329)
Comment 12: Paragraph 3.3, lines 333-334: “lower than in chondrocytes itself” I propose “lower than in cell culture supernatants of only the chondrocytes”. Response 12: The description got changed according to the suggestion (lines 348- 349).
Comment 13: Paragraph 3.4, line 346: Instead of “To measure the influence of IPFP secreted proteins”, I would say “To measure the levels of IPFP secreted proteins in the IPFP tissue”, to make things more clear. Response 13: We used the suggested phrase instead of the one existing (line 361).
Comment 14: Paragraph 3.4, line 358: “Platelet-derived growth factor showed similar like….catabolic features on the human cartilage” It is not clear how the underlined part is connected to the results presented in this manuscript. I propose to change this part of the sentence for “a trend to increase in OA IPFP”. Response 14: The sentence got changed according to the suggestion in the comment (line 373).
Comment 15: Discussion, lines 425-426: Leptin content in IPFP did not significantly differ between OA patients and controls, so the authors should mention this here. Response 15: We added that information into the phrase here (lines 444- 445).
Comment 16: Discussion, line 427: Besides the just mentioned absence of significance, this sentence is very strange. The authors write here that a possible pathogenetic reason for higher leptin in OA IPFP could be that leptin upregulates LOXL3. I think that the authors mean to say that leptin could be connected to OA via its capacity to upregulate LOXL3. Response 16: This is exactly what we tried to emphasize. We added the suggested statement instead of the existing one (lines 446- 447).
Comment 17: Discussion, line 433: PDGF-BB content in IPFP did not significantly differ between OA patients and controls, so the authors should mention this here. There is high variability between the effects of OA IPFPs of OA patients on PDGF-BB content. Response 17: The suggested statement was added (lines 458- 459) in the discussion section.
Comment 18: Discussion, lines 436-440: Besides the just mentioned absence of significance, the next sentences are very strange. Elevated NSC levels of PDGF-BB in the IPFP OA samples would be explained by the findings in a mouse experiment. I think that the authors mean to say that PDGF-BB could be connected to OA via its role in subchondral bone angiogenesis. Response 18: This is exactly what we wanted to communicate by this sentence. We changed some words within the phrase and added the recommended postulation (lines 452- 454). Comment 19: Discussion, lines 454-455: “Therefore, the IPFP has suppressed the TNF-α production of the chondrocytes. This conclusion is not valid, because both the chondrocytes and the IPFP produce TNF-α, if cultured alone. When cultured together, the level is lower, but this could also mean that chondrocytes have suppressed TNF-α production of the IPFP. Response 19: Thank you for pointing out the imprecision of this sentence. As the statement is not valid, we deleted it from the text to avoid misleading conclusions.
Comment 20: Discussion, Lines 507-514: In this concluding part of the discussion, again I would first mention what effects control and OA IPFPs have in common , and end with the effect that are significantly different in OA IPFP, because these are most relevant considering the aim of the study. Then MMPs do not need that much attention any more. Response 20: According to the recommended structure in the comment, we changed this chapter by pointing out the similarities first, then showing the differences and leaving the effect of the MMPs at this point out (lines 525- 534).
Comment 21: Discussion, lines 515-516: “These results underline the potential influence of the IPFP on human cartilage its certain role in the pathogenesis of OA, considering inflammation and remodeling processes”. Besides the strange construction of this sentence, I also have the comment that the authors should stay close to their own findings. There was no proof for a certain role of the IPFP in the manuscript, so I propose to confine this sentence to “These results underline the potential influence of the IPFP in the pathogenesis of OA, considering inflammation and remodeling processes”. Response 21: We changed the sentence as suggested above (lines 532- 534). |
||
Should you require any further information regarding our response, please do not hesitate to contact me at any time.
Sincerely,
Dr. med. Michal Jagielski
Round 2
Reviewer 2 Report (New Reviewer)
Comments and Suggestions for Authors
No other comments
This manuscript is a resubmission of an earlier submission. The following is a list of the peer review reports and author responses from that submission.
Round 1
Reviewer 1 Report
Comments and Suggestions for Authors
Reviewer 2 Report
Comments and Suggestions for Authors
The manuscript by Wisniewksa, et al. describes the effect of IPFP tissue from osteoarthritic or control samples on cultured human chondrocytes. They find effects on collagen gene expression, matrix metalloproteinase expression and cytokine expression. The study design is good and the manuscript is well written. The findings are relevant to the OA research field and provide more insight on the relevance of the IPFP.
A couple of main concerns should be addressed: 1) methodology for pooling chondrocytes from donors and their data presentation is not entirely clear, 2) how was IPFP tissue selected from OA donors to facilitate comparison with control (regional variation, location of sampling), 3) The conclusion about cytokine array differences for leptin and adiponectin should be further substantiated with e.g. ELISA measurements.
Major comments
Methods
Sample information;
Line 116: Please indicate the total number of healthy and OA donors used in this study. The mean age would also be useful in addition to the range.
Line 120-122: “First, every sample was macroscopically assessed as either OA (osteoarthritis) or CO (control; healthy) type based on the cartilage relief appearance.” The reviewer assumes that only athroscopic samples could be labelled as control; healthy, even if the macroscopic damage was not so clear? Please clarify.
Line 134-135: “Prior to their use in co-cultivation experiments cell pools consisting of either 9 OA or 5 CO chondrocyte samples were prepared.” In what figure where OA or CO chondrocytes used? This remains unclear. Also, did the authors validate, using gene or protein expression measurements, that OA or CO chondrocyte samples differed in any way?
Line 135-137: “In the result section we showed the chondrocytes as mixed categories of both OA and control samples (di-vided only in time lapses as d-2, d0 and d2) for an easier way of presentation the influence of the fat pad on them, independently to the origin of the chondrocytes.” This remains highly confusing, explain more clearly; were pools of OA or CO chondrocytes stimulated separately with IPFPs? And are the results presented as combined? In that case fact that the responses are the same would be an important finding, which is now lost in the figure presentation.
Co-culture experiments: was the viability of IPFP tissue evaluated after the culture period? It is possible that immune cells die in 0.5% FCS within a day or two. This could affect the outcome of the co-culture experiments.
RTD-PCR;
Line 155-163; please add relevant protocol information according to the MIQE guidelines (PMID: 19246619), e.g. reaction volumes and template/primer concentration, thermal cycler settings, etc.
Statistical analysis;
Line 204-208: The reviewer is well aware that graphpad statistical analysis software returns a large number of asterisks as a function of the p. value. However the meaning of a statistically significant result, defined as p. value < 0.05, does not change with more asterisks or lower p. value. In the results section: if authors want to report p-value’s, then please report exact p-value’s and remove the asterisks in the main text.
Results:
IPFP sampling;
Line 211-213: In an arthroscopic procedure, only a portion of the IPFP would be removed. Isn’t it very logical that those control samples are smaller due to a different surgical intervention? This is described as if this is a surprising and important finding.
Line 436-440: “During arthroscopies we could cut out only a very small representative piece of the IPFP from control samples, whereas IPFP from a total knee replacement was basically a side product of the whole intervention itself.”
Is the approach for athroscopy through the center of the IPFP or on the side? This may lead to sampling bias that is not present for OA samples.
“In order to make a comparison between those 2 IPFP groups possible, OA tissue was cut with a stencil (6 mm diameter) to make the size of it comparable with the Co-tissue.” Was a similar position of the IPFP sampled as in an athroscopic procedure?
See also the remark below about Line 217-222.
Line 217-222: How were regional variation and potential sample bias of the IPFP taken into account? See for example Haartmans, et al. Analytical biochemistry 2023 (PMID: 36521559) for a whole mount IPFP section with variable regions of connective and adipose tissue. A supplemental figure with H&E stainings of all donors used would aid in substantiating the conclusion of the authors, which appears to be based on N=1.
Figures & flow chart;
Figure 4A: What happened to the error bar of Col1a1? More variation is expected from 6 individual measurements.
Figures with bar graphs: to the reviewer it would make interpretation of the figure more intuitive when the control IPFP is shown first (centre) and then OA IPFP’s (on the right).
Figure 8: were there other significant findings from the 120 proteins on the array? If the authors only aimed to detect leptin and adiponectin, an ELISA would have sufficed. A supplemental list of the mean intensity’s for detected proteins could be valuable reference material for the content of IPFP secretome and future IPFP studies.
Flow chart: a table with the 3 experimental conditions and =, ↑ or ↓ would be easier to read in the opinion of the reviewer.
Discussion
Line 373-375: This is reminiscent of a recent study where OA-SF of human origin induced COL1A1 expression in human articular chondrocytes when compared to FCS by Housmans, et al. OA&C 2022 (PMID: 35176481). This reference might be useful. Did the authors evaluate cell number in response to the 2 days incubation with or without IPFP?
Line 396: “We found a higher level of leptin content in IPFP tissues from donors suffering from OA 396 than from healthy.” Can this really be concluded, since no statistically significant difference was found? Perhaps additional ELISA measurements (increasing the N) could substantiate this conclusion.
Line 410-412: “Additionally, we also found a high secretion of adiponectin in both IPFP groups (OA and healthy) in the protein array.” Same point concern as for line 396.
Minor comments
Line 54/55; are there more sources on mean implant failure? Since the authors emphasize “some authors” it appears that 5.9 years is not the consensus. The referenced study appears rather solid with ~900 patients, but this is not much given that 400.000 people receive a joint replacement per year in Germany alone.
Line 432/433: exclusion criteria should be mentioned in the methods section.
Line 456: “epi-genetical changes within their DNA” The authors appear to refer exclusively to DNA methylation, however histone modifications are also important carriers of epigenetic information, which is not contained within the DNA.
Comments on the Quality of English LanguageLine 40: was the OA of the knee à was OA of the knee?
Line 45: seem à seems?
Line 97: a period is lacking at the end of the sentence.
Line 184: a period is lacking at the end of the sentence.
Line350: “In the chosen assay is no direct cell-tissue contact.”à … there is no?
Line 431/432: “of an OA.” à of OA?
Line 455: epi-genetical changes à epigenetic changes?